# Reproduction Study of Variational Fair Clustering

## Reproducibility Summary

**Scope of Reproducibility**

Variational Fair Clustering (VFC) is a general variational fair clustering framework that is compatible with a large class of clustering algorithms, both prototype-based and graph-based (Ziko et al., 2021). VFC is capable of handling large datasets and offers a mechanism that allows for a trade-off between fairness and clustering quality. We run a series of experiments to evaluate the major claims made by the authors. Specifically, that VFC is on par with SOTA clustering objectives, that it is scalable, that it has a trade-off control, and that it is compatible with both prototype-based and graph-based clustering algorithms.

**Methodology**

To reproduce the results from Ziko et al., the original code is altered by removing bugs. This code is used to perform reproduction experiments to test the four claims made by the authors, as described above. Furthermore, three replication experiments have been implemented as well: different values for the trade-off parameter and Lipschitz constants have been investigated, an alternative dataset is used, and a kernel-based VFC framework has been derived and implemented.

**Results**

We found that that three of the four claims made by Ziko et al. are supported, and that one claim is partially supported. VFC is mostly on par with SOTA clustering objectives, if the trade-off parameter and Lipschitz constant are tuned. Additionally, we verified that VFC is scalable on large-scale datasets and found that the trade-off control works as stated by the authors. Moreover, we conclude that VFC is capable of handling both prototype-based and graph-based datasets. Regarding the replicability of VFC, the experiment on the alternative dataset did not indicate that VFC is worse than SOTA baselines. The proposed kernel-based VFC performs on par with the original framework.

**What was easy and difficult**

The original paper provides extensive theoretical derivations and explanations of the VFC approach, both through derviations and text. Moreover, the code of the original paper was publicly available. The original authors responded quickly to our mails and were very willing to discuss our results.

Although the VFC code was publicly available, it was undocumented and contained some bugs that were hard to find given the lack of documentation. Moreover, there were vast differences between the implementation of the original authors and the baseline models. This required conversions between the models for the comparisons. Lastly, running the VFC code took many hours, which resulted in us not being able to run all algorithm-dataset combinations we wanted to.

**Communication with original authors**

The original authors have been approached twice. The mail contact helped clarify implementation details, particularly regarding the Ncut algorithm. The authors explained and specified the usage of the trade-off parameter and the Lipschitz constant. Additionally, they explained how they obtained the $K$-means baseline results. The authors have been informed about our proposed kernel-based VFC framework and replied with enthusiasm.

Submitted to ML Reproducibility Challenge 2020. Do not distribute.

# 1 Introduction

Fairness in machine learning (ML) has received significant interest as ML algorithms are used in, for example, financial, marketing, and educational decision purposes, thereby directly influencing human lives. However, achieving fairness is still a challenge due to neglected or unaware biases in the data and ambiguity of the definition (Mehrabi et al., 2021).

One of the notions of fairness is *fair clustering* (Chierichetti et al., 2018; Bera et al., 2019; Backurs et al., 2019; Huang et al., 2019; Rösner and Schmidt, 2018; Schmidt et al., 2018; Kleindessner et al., 2019). Fair clustering is a clustering approach where the resulting cluster assignment should not be disproportionately different for individuals with different protected attributes (e.g. gender). This is achieved by balancing the distribution of protected subgroups in each cluster. A limitation of state-of-the-art (SOTA) fair clustering algorithms is that they can only be used for either prototype-based or graph-based objectives. For large datasets, graph-based clustering algorithms pose additional difficulties since they are not computationally scalable.

In the paper *Variational Fair Clustering* (VFC), (Ziko et al., 2021) address these problems. They propose the VFC framework that provides a general fair formulation for both prototype-based and graph-based clustering objectives by incorporating an original fairness term. This framework is implemented using three well-known clustering objectives ($K$-medians, $K$-means, and Ncut), and are compared to their respective SOTA versions from (Backurs et al., 2019), (Bera et al., 2019), and (Kleindessner et al., 2019).

# 2 Scope of reproducibility

In this reproducibility study, we focus on the main claims of Ziko et al. (original authors, OA) stated in their paper (original paper, OP). The SOTA fair clustering algorithms are referred to as baselines. The main claims of the OP are:

Claim 1 VFC is on par with state-of-the-art clustering objectives on the Synthetic, Synthetic-unequal, Adult, Bank, and Census II datasets:

    a VFC using $K$-medians has lower objective energies, lower fairness errors, and higher balances than the baseline (Backurs et al., 2019).

    b VFC using $K$-means has lower objective energies than the baseline (Bera et al., 2019), but will achieve similar fairness errors and balances.

    c VFC using Ncut has slightly higher objective energies than the baseline (Kleindessner et al., 2019), but achieves similar fairness errors and balances[1].

Claim 2 It is computationally feasible to run VFC using the Ncut algorithm on large-scale datasets that have 2.5 million records.

Claim 3 VFC provides the best clustering objective with the smallest $\lambda$ that satisfies a pre-defined fairness level $\sum_k \mathcal{D}_{KL}(U||P_k) \leq \epsilon$.

Claim 4 VFC is capable of performing both prototype-based and graph-based clustering objectives.

# 3 Methodology

We test the validity of the claims using the provided VFC framework. In the following sections we cover the description of this architecture, the used datasets and hyperparameters. We include an experimental setup and code section which covers three reproduction experiments and three replication exerpiments. The former is performed to evaluate the reported results, and the latter is conducted to further analyse the claims and improve on the proposed framework. This work includes tuning hyperparameters, testing alternative datasets, and introducing a kernel-based clustering approach.

## 3.1 Model description

The VFC objective is described by a variational trade-off between a clustering objective and an original fairness objective. The fairness objective is given by the KL-divergences between the demographic proportions in the data and the distributions of each cluster. The trade-off is regulated by the hyperparameter $\lambda$. The OA derive a general convex-concave formulation for the VFC objective, which is optimised using auxiliary functions (for the formal details, see the OP). The VFC requires a predefined number of clusters ($K$), a trade-off parameter ($\lambda$), and a sensitive attribute (e.g. gender).

### 3.2 Datasets

The VFC algorithm has been tested on five datasets, two of which were synthetically created by the OA. Both synthetic datasets have two demographic groups and 400 numeric data points. The *Synthetic* (SB) dataset is balanced, as both demographic groups contain 200 data points. The *Synthetic-unequal* (SU) dataset has 300 and 100 data points in the groups respectively. The OA also used three real datasets from the UCI machine learning repository[2]: *Bank*[3], *Adult*[4], and *Census II*[5]. We use the same sensitive attributes and remove the same values within the datasets as the OP.

Our replicability research uses the VFC algorithm on two datasets: *Student*[6] and *Drugnet*[7]. We use the sex or gender as sensitive attribute. An overview of the characteristics of the real datasets can be found in Table 4, Appendix A.

### 3.3 Hyperparameters

In the reproduction experiments, the hyperparameters are set to have the same value as in the OP. An overview of the hyperparameters can be found in Table 5, Appendix B. Note that different Lipschitz constants are used for different parts of reproduction, as is explained in more detail in section 5. The OP conducted a hyperparameter search on the $\lambda$ and $K$ parameters as two of their experiments, but did not draw any conclusions with regard to the value of $K$.

For our additional experiments, we perform hyperparameter searches for the $\lambda$ parameter and Lipschitz constant on all datasets but the Census II dataset to improve the OP's results. We considered seven different Lipschitz constants, as shown in Figure 2, with 10 different seeds each. We conducted another manual search for the $\lambda$ parameter after fixing the Lipschitz constants. Here we tested the integer values 1 up to 10, in addition to 0.5 and 1.2 for Ncut. For $K$-means and $K$-medians 20 different values between 3000 and 10000 were tested. The tuned hyperparameters are reported in Table 9, Appendix E. For the kernel approach, a small manual search has been conducted over the integers 1 up to 10 for every parameter in the kernel.

### 3.4 Experimental setup and code

The code provided by the OA[8] was used to reproduce the experiments. This code contained some bugs, hence we created an updated codebase[9] for our experiments. The code for the $K$-means baseline[10] (Bera et al., 2019) is used to create replication results. Due to limited time and resources, we have not implemented the baselines for $K$-medians (Backurs et al., 2019) and Ncut (Kleindessner et al., 2019). Lastly, the results on the Drugnet dataset obtained by Kleindessner et al. are used as baseline in one of the replication experiments.

We conduct a total of three reproduction experiments and three replication experiments. The reproduction experiments consist of comparing the baseline models to VFC, testing the scalability of the Ncut algorithm, and recreating the $\lambda$ plots from the OP. The replication experiments include tuning the Lipschitz constant, exploring the generalisability of VFC on other datasets, and introducing a kernel-based VFC.

**Reproduction experiments**    For the comparison experiments defined in Claim 1, the clustering algorithms $K$-medians, $K$-means, and Ncut are applied to all five datasets used in the OP. The performance of the algorithms is measured with three metrics as defined in the OP: clustering energy (objective), fairness error, and balance. Every algorithm-dataset combination is run with different seeds to obtain a mean and a standard deviation. All combinations are run with 30 different seeds except the Census II dataset and the Ncut algorithm, as these combinations take infeasibly long. These are run with only five different seeds. To prevent the metrics from taking outliers into account, Chauvenet's criterion (Lin and Sherman, 2007) is used (see Appendix C). We consider a statistic reproducible if is at least as good as the one reported results in the OP. Moreover, a result is unreproducible if the reproduction attempt is at least one standard deviation worse. All values in between are labelled inconsistent.

Scalability, as defined in Claim 2, is evident from the results of the fair Ncut algorithm on the largest dataset, Census II. If the OP's results of this combination are successfully reproduced in a reasonable time frame, this implies scalability.

To test Claim 3, the $\lambda$ plots in the OP's Figure 2 are reproduced by running the Fair $K$-means and Ncut algorithms on the Adult and Bank datasets with varying $\lambda$ values. The Ncut plots are generated with both a Lipschitz constant of 2.0 and 0.001.

Albeit the OA discuss Figure 3 in the OP, no claim has been made on the impact of the value of $K$ for the algorithms. Therefore, this figure is not reproduced in this research.

After failing to reproduce some reported results, it was established in communication with the OA that some reported $\lambda$ values are incorrect and therefore unknown. A manual search is executed to test 10 $\lambda$ values ranging from 100 to 1500 for the $K$-medians and $K$-means algorithms on the SB dataset. The search is not feasible for Ncut algorithm on Census II, given the limited time and resources of this research. Note that the results for $\lambda = 0$ can be interpreted as the performance of the algorithm not taking into account fairness at all.

**Tuning the Lipschitz constant**  The effect of the unreported Lipschitz constant is investigated by running the Adult and Bank datasets with seven different Lipschitz constants ranging from $10^{-5}$ to 2.0, with 10 different seeds for all three algorithms. Afterwards, 30 different seeds are tested for the tuned Lipschitz and $\lambda$ values for the Adult and Bank datasets. Lastly, we run the Ncut algorithm on Census II three times with Lipschitz $10^{-5}$ to retest scalability.

**Exploring other datasets**  To evaluate the performance of the VFC framework, the experiment performed by the OA has been replicated. The implementation[10] of the $K$-means baseline paper (Bera et al., 2019) used IBM's `Cplex Optimiser`[11], which we were not able to get full access to. Given that we were therefore limited to using smaller datasets, only the Student dataset was used. This baseline uses a fairness trade-off parameter $\delta$ describing how loose the fairness condition is, with $0 \leq \delta \leq 1$. The fairness condition is met exactly when $\delta = 0$, and is ignored when $\delta = 1$. No $\delta$ has been specified in the work of the OA; we opted for a relatively high value of $\delta = 0.9$ as a result of a small manual search: due to the small size of the dataset, all $\delta$ values up to 0.9 yielded an equal fairness error. We therefore opted to use the highest value to yield the best fairness error to favour the baseline.

Furthermore, to verify whether the VFC-algorithm can be applied to graph-based structures, as stated in Claim 4, the Fair Ncut algorithm is run on the Drugnet dataset. In the OP, the Ncut algorithm was only used on non-graphical datasets that were converted to graphical data using pair-wise affinities. In the derivation of the respective auxiliary function, the OA assume the adjacency matrix to be positive semi-definite. Hence, we evaluate if the graph-based framework also works on non-synthetic adjacencies. This is evaluated by using the Drugnet dataset and comparing it to the Ncut baseline (Kleindessner et al., 2019).

**Kernel-based clustering**  To derive a kernel-based VFC framework, a reformulated objective and corresponding auxiliary function have to be derived. Kernel-based clustering can be seen as a generalisation of $K$-means clustering. Rather than minimising the Euclidean distance between the individual points and their corresponding cluster centres, a kernel-based distance metric is minimised, i.e. the objective is given by:

$$\min_{\mathbf{S}} \sum_k \sum_p s_{p,k} d(\mathbf{x}_p, \mathbf{c}_k) \ \text{ s.t. } \ \mathbf{s}_p \in \nabla_K \forall p, \tag{1}$$

for some kernel-based distance metric $d$. This definition provides a general formulation which combined with the VFC fairness term is refereed to as kernel-based VFC. We make use of the following fact:

**Proposition 1.** *Given current clustering solution $\mathbf{S}^i$ at iteration $i$, the auxiliary function for kernel-based clustering can be written in the following general form:*

$$\mathcal{H}_i(\mathbf{S}) = \sum_p \boldsymbol{s}_p^t \boldsymbol{a}_p^i,$$

*where $\boldsymbol{a}_p^i$ is given by a kernel-based distance metric $d(\boldsymbol{x}_p, \boldsymbol{c}_k^i)$ (proof in Appendix D).*

We conduct a third experiment to evaluate the effect of using a kernel in VFC. The kernel-based approach is implemented and evaluated with the polynomial kernel, the Gaussian kernel, and the hyperbolic tangent kernel. These decisions are motivated in Appendix F. Given that no cluster labels are available, a measure of consistency within clusters, called silhouette coefficient (SC), is used as a measure, c.f. (Dinh et al., 2019). The fairness error and balance metric are used as well. To ensure that SC is not biased to any of the two algorithms, the cosine similarity is used for comparison. The use of a kernel implies a computational complexity of $\mathcal{O}(N^2KM)$ and hence only the smaller datasets such as SB, SU, and 2,500 random entries from the Bank dataset are considered (more information on the complexity is given in Appendix F). Due to time constraints, only a single run has been done for each dataset and kernel combination.

### 3.5  Computational requirements

All results were obtained on Windows 10 with an Intel i7-10875H CPU. The GPU is not used as the algorithms are optimised for CPU multiprocessing. In total, the runtime of all reported results in this paper was 107 hours. Including

| | Datasets | Objective | | Fairness error / Balance | |
|---|---|---|---|---|---|
| | | Baseline | VFC | Baseline | VFC |
| **F $K$-medians** | SB | 299.86 | **289.08** (±2.03) | **0.0 / 1.0** | 0.82 (±1.05) / 0.34 (±0.21) |
| | SU | 185.47 | **174.82** (±0.00) | 0.77 / 0.21 | **0.0** (±0.00) / **0.33** (±0.00) |
| | Adult | 19330.93 | **17887.87** (±307.59) | 0.27 / 0.31 | **0.01** (±0.00) / **0.42** (±0.01) |
| | Bank | N/A | **20242.38** (±403.62) | N/A / N/A | **0.04** (±0.00) / **0.17** (±0.00) |
| | Census II | 2385997.92 | **1746911.27** (±10270.47) | 0.41 / 0.38 | **0.02** (±0.00) / **0.75** (±0.04) |
| **F $K$-means** | SB | 758.43 | **203.66** (±2.55) | **0.0 / 1.0** | 2.43 (±1.47) / 0.27 (±0.44) |
| | SU | 180.0 | **159.75** (±0.00) | **0.0 / 0.33** | **0.0** (±0.00) / **0.33** (±0.00) |
| | Adult | 10913.84 | **10355.98** (±328.43) | 0.02 / **0.41** | **0.01** (±0.00) / 0.4 (±0.01) |
| | Bank | 11331.51 | **9907.19** (±550.52) | **0.03** / 0.16 | 0.08 (±0.00) / **0.17** (±0.00) |
| | Census II | **1355457.02** | 2279984.75 (±1548556.61) | **0.07 / 0.77** | 41.86 (±51.83) / 0.42 (±0.34) |
| **F Ncut** | SB | **0.0** | 0.2 (±0.10) | **0.0 / 1.0** | **0.0** (±0.00) / 0.98 (±0.02) |
| | SU | 0.03 | **0.02** (±0.03) | **0.0 / 0.33** | **0.0** (±0.00) / 0.32 (±0.01) |
| | Adult | **0.47** | 0.78 (±0.02) | **0.06** / 0.32 | 0.08 (±0.02) / **0.36** (±0.03) |
| | Bank | N/A | **0.65** (±0.01) | N/A / N/A | **0.25** (±0.03) / **0.14** (±0.01) |
| | Census II | N/A | **1.74** (±0.14) | N/A / N/A | **21.84** (±10.02) / **0.0** (±0.00) |

Table 1: Comparison of the proposed Fair algorithms to baseline models

explorative experiments, the total runtime was 160 hours. Of this total, 135 hours were for conducting the experiments on the Census II dataset. Specific runtimes per algorithm-dataset combination are given in Table 11, Appendix E.

# 4   Results

We have conducted the aforementioned experiments and gathered the results together in the following two subsections. The first subsection focuses on the findings of the reproduction experiments. The second subsection covers the findings of the replication experiments.

## 4.1   Results reproducing original paper

The results of the reproduction experiments are reported in their respective columns in Table 1. The mean is reported, and the standard deviation is given between parentheses. The bold numbers indicate the best values for a given dataset. Table 6, Table 7, and Table 8 in Appendix E visualise the comparison of the OP's results to ours using the parameters listed in Table 5. The red entries indicate results that were unreproducible with the original hyperparameters, and the orange entries correspond to those that were inconsistent.

**Comparison between Backurs et al. and Fair K-medians**   In total, 10 out of 12 results support Claim 1a with the initial hyperparameters. The fairness of the SB dataset is labelled as an inconsistent reproduction, and the balance is labelled as an unreproducible result. Tuning the $\lambda$ improved the fairness, but the balance did still not reach baseline performance (Table 2). Given that the vast majority of results lie well within reproduction range, and that the only deviating result is on a small synthetic dataset, Claim 1a is almost completely supported.

**Comparison between Bera et al. and Fair K-means**   The fairness error and balance for the SB dataset could only be reproduced after tuning $\lambda$ (Table 2). Furthermore, reproduction of the fairness error and balance on the SU dataset was only achieved after the exclusion of bad seeds using Chauvenet's criterion (Appendix C). The seeds for all excluded outliers are shown in Table 10, Appendix E. All metrics for the Census II dataset deviated from the OP and were worse than the baseline results. However, three out of five runs achieved similar results to the OP. In this case, the two bad seeds were not flagged as outliers. Claim 1b is therefore also mostly supported, but less so than in the OP.

**Comparison between Kleindessner et al. and Fair Ncut**   The reproduction results of the Ncut algorithm show similar performance compared to the baseline model. Note that, in our results, the Ncut algorithm performed better than the baseline in terms of the objective on the SU dataset. Regarding fairness, the reproduced VFC is on par with the baseline for both synthetic datasets, but worse for the Adult dataset. Moreover, the reproduced balance is only better for the Adult dataset. Thus, Claim 1c is also mostly supported.

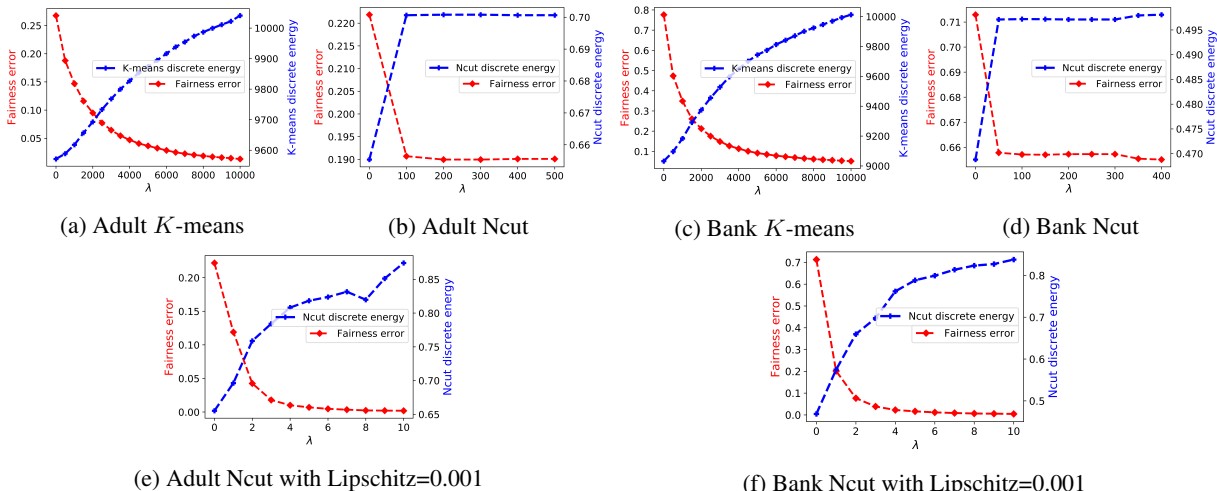

Figure 1: Reproduction of fairness error across different $\lambda$ values and datasets

| Algorithms | SB dataset | | | |
| --- | --- | --- | --- | --- |
| | Objective | | Fairness error / Balance | |
| | baseline | $\lambda$ tuned | baseline | $\lambda$ tuned |
| $K$-medians | **299.86** | 314.98($\pm$43.23) | **0.0 / 1.0** | **0.0**($\pm$0.00) **/ 0.93**($\pm$0.05) |
| $K$-means | 758.43 | **207.8**($\pm$0.00) | **0.0 / 1.0** | **0.0**($\pm$0.00) **/ 1.0**($\pm$0.00) |

Table 2: Comparison of the proposed Fair $K$-medians and $K$-means to the baselines ((Backurs et al., 2019) and (Bera et al., 2019), respectively) on the SB dataset with tuned $\lambda$ values.

**Scalability**   The results for the Fair Ncut algorithm were not successfully reproduced for the Census II dataset using the reported hyperparameters in the OP. Adjusting the Lipschitz constant to 1.0, as suggested by the authors, did not solve this issue. Lowering the Lipschitz constant to $10^{-5}$ did enable convergence, taking 34 hours per run on average. This achieved reasonable results, despite not reaching the reported performance in the OP. Thus, Claim 2 is supported, but less so than in the OP.

$\lambda$ **plots**   In Figure 1, the blue curve depicts the discrete-valued clustering objective ($K$-means or Ncut) obtained at convergence as a function of $\lambda$. The fairness error is denoted in red. As shown, increasing the $\lambda$ parameter lowers the fairness error. Unlike the OP's reported result of the $K$-means objective for the Adult dataset, Figure 1a does not show the oscillating behaviour. Different from Figure 2 of the OP, the Lipschitz constant was set to 0.001 for the Ncut plots in Figure 1e and Figure 1f to improve convergence. This is reflected in the lower fairness errors. By choosing a $\lambda$ arbitrarily large, an arbitrarily small fairness error will be found as seen in 1. Hence, Claim 3 is supported.

### 4.2   Results beyond original paper

**Tuning the Lipschitz constant**   Figure 2 shows that Lipschitz constants down to $10^{-5}$ speed up convergence for $K$-means and Ncut on the Bank dataset. These results are reflected for all three algorithms in both Synthetic datasets and the Adult dataset. This is also shown in the runtimes in Table 11, Appendix E. Further decreasing the Lipschitz constant leads to invalid results; `NaN` values impede convergence.

In all cases, the reproduced results with the original hyperparameters were equal to, or improved by, lower Lipschitz constants aside from the balance of $K$-medians on the Adult dataset, and the fairness error of $K$-medians on the Bank dataset (Table 3). However, the main improvement lies in the convergence.

**Other datasets**   For the Student dataset, the baseline algorithm resulted in a clustering objective of 128.012, a fairness error of 0.0056, and a balance of 0.8327. Additionally, the VFC using $K$-means and $\lambda = 50$ gave an objective of 341.892, a fairness error of 0.0032, and a balance of 0.8982, therefore having a higher objective, lower fairness error, and higher balance.

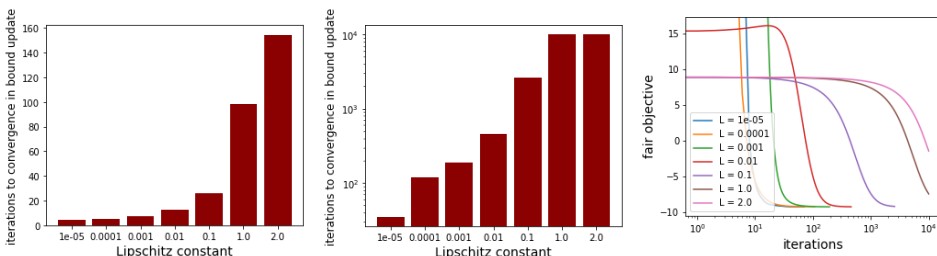

Figure 2: Convergence iterations of a VFC bound update with different Lipschitz constants on the Bank dataset. On the left the convergence of a $K$-means bound update is displayed. The middle and right plots display convergence of Ncut, where the right plot shows the fair objective by iteration.

| Dataset | Objective | | Fairness error / Balance | |
|---|---|---|---|---|
| | VFC | VFC lower Lipschitz | VFC | VFC lower Lipschitz |
| **Fair $K$-medians** | | | | |
| Adult | 17887.87($\pm$307.59) | **17513.1**($\pm$182.47) | **0.01**($\pm$0.00) / **0.42**($\pm$0.01) | **0.01**($\pm$0.00) / 0.4($\pm$0.00) |
| Bank | 20242.38($\pm$403.62) | **19743.99**($\pm$341.91) | **0.04**($\pm$0.00) / **0.17**($\pm$0.00) | 0.05($\pm$0.01) / **0.17**($\pm$0.00) |
| **Fair $K$-means** | | | | |
| Adult | 10355.98($\pm$328.43) | **10103.2**($\pm$130.60) | **0.01**($\pm$0.00) / **0.4**($\pm$0.01) | **0.01**($\pm$0.00) / **0.4**($\pm$0.00) |
| Bank | 9907.19($\pm$550.52) | **9533.59**($\pm$188.53) | 0.08($\pm$0.00) / **0.17**($\pm$0.00) | **0.06**($\pm$0.00) / **0.17**($\pm$0.00) |
| **Fair Ncut** | | | | |
| Adult | 0.78($\pm$0.02) | **0.77**($\pm$0.04) | 0.08($\pm$0.02) / 0.36($\pm$0.03) | **0.05**($\pm$0.01) / **0.37**($\pm$0.01) |
| Bank | 0.65($\pm$0.01) | **0.56**($\pm$0.06) | 0.25($\pm$0.03) / **0.14**($\pm$0.01) | **0.14**($\pm$0.02) / **0.14**($\pm$0.01) |
| Census II | 1.74($\pm$0.14) | **1.33**($\pm$0.00) | 21.84($\pm$10.02) / 0.0($\pm$0.00) | **0.4**($\pm$0.00) / **0.47**($\pm$0.00) |

Table 3: Comparison of the reproduced VFC results to the experiments with lower Lipschitz constants

Furthermore, running the VFC Fair Ncut algorithm on the Drugnet dataset resulted in an objective score of 0, a fairness error of 0.06, and a balance of 0.24. The baseline results are not exactly reported in (Kleindessner et al., 2019). However, the objective and balance can be interpreted from their Figure 5, which approximately equals an objective of 0.01, and a balance of 0.26. Thus, VFC obtains a lower objective and lower balance. Interestingly, the obtained objective score of 0 indicates that the optimal Ncut solution has been found. Hence, we can conclude that Claim 4 is supported.

**Kernel-based VFC** The kernel-based VFC obtains the same silhouette coefficients, fairness error, and balance as the standard VFC. The results are summarised in Table 13 in Appendix F.

## 5 Discussion

Given the results shown in section 4 and the varying outcomes contrasting the results in the OP, we conclude that not all claims presented in section 2 are supported.

**Reproduction** Based on our results using the original hyperparameters, Claim 1 cannot be supported. We therefore discuss the validity of Claim 1 based on the tuned $\lambda$ values. Running $K$-medians on the SB dataset yielded a similar objective, but a dissimilar fairness error and balance. As expected, increasing the $\lambda$ parameter did improve the fairness error and balance, but did so at the cost of the clustering quality as suggested in Figure 1 and Table 2. For the other datasets, we were able to find hyperparameters that made the VFC framework compatible with the baseline, and hence Claim 1a is mostly supported.

Surprisingly, the results for $K$-means on the SB dataset did improve with a tuned $\lambda$ parameter. The similarity between the results on the SU dataset was achieved after the removal of bad seeds. The $K$-means algorithm performed differently on the Census II dataset than reported by the OA. Due to time constraints, we were not able to explore this further. Given these judgements, we conclude that Claim 1b is also mostly supported.

The reproduction results of the Ncut algorithm show that the SU dataset had a better clustering objective. The algorithm also performed better on the Adult dataset as the balance was higher. As mentioned in the OP, there are no baseline

results that can be compared to our results for the Bank and Census II datasets. Hence, we compare these reproduced results only to the results obtained in the OP. For the Bank dataset, the objective worsened, but the fairness and balance were improved. The results on the Census II dataset are not as reliable, since we ran the experiment a total of five times. All in all, Claim 1c is therefore partially supported.

We have tuned the Lipschitz constant such that it did not return `NaN` values on the Census II dataset while using the Ncut algorithm. It was not feasible to run many experiments, as a Lipschitz constant of $10^{-5}$ took an average of 34 hours to converge. Thus, we observe that VFC using the Ncut algorithm scales to large datasets. Although our results did not exactly reflect those of the OP, the performance was still reasonable. Hence, Claim 2 is verified.

As mentioned earlier, there is a trade-off between the clustering objective and the fairness. Figure 1 shows that, when the $\lambda$ increases, the clustering objective increases and the fairness decreases. Observe that we do not have the oscillating behaviour as in the OP for the $K$-means algorithm on the Adult dataset, whichmay be caused by the seed that was used to run this experiment. Unfortunately, we have not explored this further due to limited time. Thus, Claim 3 is supported.

**Replication**  We found that decreasing the Lipschitz-constant to $10^{-5}$ improved the convergence speed, and in some instances performance. The proposed VFC algorithm does not perform worse than the $K$-means baseline. The $K$-means and $K$-medians experiments have shown that VFC is capable of performing prototype-based clustering objectives. The Drugnet dataset, combined with the Ncut algorithm, has shown that VFC can perform graph-based clustering objectives as well, and that it is on par with its baseline. Hence, Claim 4 is supported.

The results obtained with the kernel-based VFC were in-line with those found using the formulation of the OP, suggesting that the kernel-based approach finds the same solutions. Due to limited time, no extensive parameter search has been done. Better parameters could improve the expressiveness of the kernels, potentially leading to better results.

**What was easy and difficult**  The original paper is well-structured and contains elaborate theoretical derivations and explanations, which made the concept of the VFC easier to understand. During the reproduction of the experiments, the provided code from the original paper was greatly beneficial. The OA responded quickly and were very willing to help. Despite the access to the original code, it was initially challenging to use as it was undocumented. Another problem was that the OA used a different Lipschitz constant than was used in the code. This issue was found later, after communicating with the OA. However, it is still unknown which Lipschitz constant the OA used for their experiments. Next to that, the code to obtain the results for the baseline models was missing as well. This was necessary for the replication experiment of the Student dataset. The implementation of the $K$-means baseline model was publicly available, but the metric used for clustering differed from those the OA used, which made comparison difficult. Hence, the results of the baseline needed to be converted into the measures that were used by the OA. Moreover, the $K$-means baseline could not be implemented on large datasets, as there was no access to IBM's premium `Cplex Optimiser`.

**Shortcomings of the original paper**  The shortcomings found in the OP are: correctness of the reported $\lambda$ parameters, not stating the correct Lipschitz value, and the errors in the provided code.

**Conclusion**  This report shows both a reproduction and a replication of *Variational Fair Clustering*. We conclude that solely the OP and the provided code do not suffice to validate the claims stated by the OA. Investigating more large-scale datasets and the effects of other Lipschitz constants are suggested for future research. Potentially, a Lipschitz constant can be found that provides rigidness and fast convergence. This paper introduced the notion of a flexible kernel-based approach. As its results are already on par with the VFC framework proposed by the OA, this approach looks promising. Further investigation by using different kernels, or improving parameters, may be beneficial.

Unfortunately, due to limited time and resources, other aspects of the VFC framework were not examined. Investigating different fairness metrics and studying the influence of larger $K$ values on the clustering energy, fairness and balance, may improve the VFC performance.

All in all, the VFC framework allows for a large class of clustering algorithms to be used in fair clustering. The framework is capable of handling large datasets and offers a mechanism that allows for a trade-off between fairness and clustering quality. Moreover, the resulting algorithms are competitive with SOTA fair clustering algorithms.

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

## A Datasets

| Dataset | Description | Original of datapoints | Sensitive attribute | Demographic groups | Preprocessing | of datapoints postprocessing | Other attributes |
|---------|-------------|------------------------|---------------------|--------------------|--------------|------------------------------|------------------|
| Bank | Marketing campaigns of a Portuguese banking institution corresponding to each client | 41,188 | Marital status | single, married, divorced, unknown | Removing unknown | 41,108 | 6 categorical, 4 binary, 6 numerical |
| Adult | US census record dataset from 1994 | 32,561 | Gender | female, male | N/A | N/A | 6 numerical, 7 categorical, 2 binary |
| Census II | Large scale US census record data from 1990 | 2,458,285 | Gender | female, male | N/A | N/A | 67 other attributes |
| Student | Achievements in Portuguese schools on the subject of mathematics | 395 | Sex | female, male | N/A | N/A | 6 other attributes |
| Drugnet | Habits of drug users in Hartford. The edges represent acquaintance-ship. (graph dataset) | 293 | Gender | female, male | N/A | N/A | 1 numerical |

Table 4: Characteristics of datasets

## B Hyperparameters

| | | Dataset | | | | |
|---|---|---|---|---|---|---|
| | | Synthetic | Synthetic-unequal | Adult | Bank | Census II |
| | Lipschitz | $\lambda$ | | | | |
| $K$-medians | 2.0 | 10 (600) | 10 | 9000 | 9000 | 500000 |
| $K$-means | 2.0 | 10 (100) | 10 | 9000 | 6000 | 500000 |
| Ncut | 1.0/2.0 | 10 | 10 | 10 | 40 | 100 |

The Ncut shows two Lipschitz constants because different values are used for reproduction: Table 1 & Figure 1

The $\lambda$ value in parentheses indicate the tuned lambda found in section 4.2.

Table 5: Hyperparameters

## C Outlier detection (Chauvenet's criterion)

The idea behind Chauvenet's criterion is similar to standard hypothesis testing using $Z$-scores. The aim is to find the corresponding standard normal distribution to the distribution that is being considered and see if the relevant point is reasonably close to the mean. The Chauvenet's maximum allowed deviation for a dataset of size $N$, denoted as $T_N$, is given by the the inverse of the standard normal cumulative distribution in $\frac{1}{4N}$, and therefore a data point $\mathbf{x}_p$ is considered an outlier if

$$\frac{|\mathbf{x}_p - \bar{\mathbf{x}}|}{s_x} > T_N,$$

where $\bar{\mathbf{x}}$ denotes the sample mean, $s_x$ denotes the sample standard deviation.

# D  Proofs

**Proof of Proposition 1**

*Proof.* As seen in Equation 1, the $K$-means objective can be altered to

$$\min_{\mathbf{S}} \sum_k \sum_p s_{p,k} d(\mathbf{x}_p, \mathbf{c}_k) \ \text{s.t.} \ \mathbf{s}_p \in \nabla_K \forall p,$$

where $d$ is some kernel-based distance metric. Let $\kappa : X \times X \to \mathbb{R}$ be an arbitrary kernel function. We can define the following distance-based metric based on $\kappa$ (Hall, 2012):

$$d(\mathbf{x}_p, \mathbf{c}_k) = \kappa(\mathbf{x}_p, \mathbf{x}_p) - 2\frac{\sum_{l=1}^n s_{l,k}\kappa(\mathbf{x}_l, \mathbf{x}_p)}{\sum_{l=1}^n s_{l,k}} + \frac{\sum_{q=1}^n \sum_{l=1}^n s_{q,k}s_{l,k}\kappa(\mathbf{x}_q, \mathbf{x}_l)}{(\sum_{q=1}^n s_{q,k})^2}. \tag{2}$$

By a similar argument as given in Proposition 2 of the OP, it follows that

$$\sum_p s_{p,k}(\mathbf{x}_p - \mathbf{c}_k)^2 \leq \sum_p s_{p,k} d(\mathbf{x}_p - \mathbf{c}_k^i).$$

hence, the auxiliary function for Kernel-Based Clustering can be written as

$$\mathcal{H}_i(\mathbf{S}) = \sum_p \mathbf{s}_p^t \mathbf{a}_p^i,$$

where $a_{p,k}^i$ is given by $d(\mathbf{x}_p, \mathbf{c}_k^i)$ as given in Equation 2. $\qquad\square$

# E   Experimental results

**Comparison between OP and reproduction results**

| Dataset | Fair $K$-medians | | | |
| | Objective | | Fairness error / Balance | |
| | Original | Reproduction | Original | Reproduction |
| --- | --- | --- | --- | --- |
| Synthetic | 292.40 | 289.08($\pm$2.03) | 0.00/ 1.00 | 0.82($\pm$1.05) / 0.34($\pm$0.21) |
| Synthetic-unequal | 174.81 | 174.82($\pm$0.00) | 0.00 / 0.33 | 0.00($\pm$0.00) / 0.33($\pm$0.00) |
| Adult | 18467.75 | 17887.87($\pm$307.59) | 0.01 / 0.43 | 0.01($\pm$0.00) / 0.42($\pm$0.01) |
| Bank | 19527.08 | 20242.38($\pm$403.62) | 0.02/ 0.18 | 0.04($\pm$0.00) / 0.17($\pm$0.00) |
| Census II | 1754109.46 | 1746911.27($\pm$10270.47) | 0.02 / 0.78 | 0.02($\pm$0.00) / 0.75($\pm$0.04) |

Table 6: Comparison of Fair $K$-medians results by the original paper to this paper's reproduction

| Dataset | Fair $K$-means | | | |
| | Objective | | Fairness error / Balance | |
| | Original | Reproduction | Original | Reproduction |
| --- | --- | --- | --- | --- |
| Synthetic | 207.80 | 203.66($\pm$2.55) | 0.00/ 1.00 | 2.43($\pm$1.47) / 0.27($\pm$0.44) |
| Synthetic-unequal | 159.75 | 159.75($\pm$0.00) | 0.00 / 0.33 | 0.00($\pm$0.00) / 0.33($\pm$0.00) |
| Adult | 9984.01 | 10355.98($\pm$328.43) | 0.02 / 0.41 | 0.01($\pm$0.00) / 0.40($\pm$0.01) |
| Bank | 9392.20 | 9907.19($\pm$550.52) | 0.05/ 0.17 | 0.08($\pm$0.00) / 0.17($\pm$0.00) |
| Census II | 1018996.53 | 2279984.75($\pm$1548556.61) | 0.02/ 0.78 | 41.86($\pm$51.83) / 0.42($\pm$0.34) |

Table 7: Comparison of Fair $K$-means results by the original paper to this paper's reproduction

| Dataset | Fair Ncut | | | |
| | Objective | | Fairness error / Balance | |
| | Original | Reproduction | Original | Reproduction |
| --- | --- | --- | --- | --- |
| Synthetic | 0.00 | 0.20($\pm$0.10) | 0.00 / 1.00 | 0.00($\pm$0.00) / 0.98($\pm$0.02) |
| Synthetic-unequal | 0.06 | 0.02($\pm$0.03) | 0.00 / 0.33 | 0.00($\pm$0.00) / 0.32($\pm$0.01) |
| Adult | 0.74 | 0.78($\pm$0.02) | 0.08 / 0.30 | 0.08($\pm$0.02) / 0.36($\pm$0.03) |
| Bank | 0.58 | 0.65($\pm$0.01) | 0.39 / 0.14 | 0.25($\pm$0.03) / 0.14($\pm$0.01) |
| Census II | 0.52 | 1.74($\pm$0.14) | 0.41/ 0.43 | 21.84($\pm$10.02) / 0.00($\pm$0.00) |

Table 8: Comparison of Fair Ncut results by the original paper to this paper's reproduction

**Tuned Lipschitz hyperparameters**

| Dataset | Fair $K$-medians | | Fair $K$-means | | Fair Ncut | |
| | $\lambda$ | Lipschitz | $\lambda$ | Lipschitz | $\lambda$ | Lipschitz |
| --- | --- | --- | --- | --- | --- | --- |
| Adult | 6500 | 0.5 | 9500 | 1.0 | 2 | 1e-5 |
| Bank | 9000 | 1.0 | 9500 | 1.0 | 1.2 | 0.0001 |
| Census II | N/A | N/A | N/A | N/A | 0.5 | 1e-5 |

Table 9: Lipschitz / $\lambda$ combinations for the results in Table 3

## Excluded outlier seeds

|                    | Outliers          |                |            |
|--------------------|-------------------|----------------|------------|
| **Datasets**       | $K$-medians       | $K$-means      | Ncut       |
| Synthetic          | N/A               | N/A            | N/A        |
| Synthetic-unequal  | N/A               | 20, 22, 24     | 20, 22, 24 |
| Adult              | 5                 | 5              | N/A        |
| Bank               | N/A               | N/A            | 2          |
| Census II          | 4                 | N/A            | N/A        |

Table 10: Seeds of the outliers that were excluded from analysis per algorithm-dataset combination

## Runtime

|                   | Runtime in seconds | | | |
|-------------------|-------------------------------|------------------------------|------------------------------------------|------------------------------------|
| **Datasets**      | $K$-medians                   | $K$-means                    | Ncut                                     | Ncut Lipschitz                     |
| Synthetic         | $4.8(\pm1.7) \times 30$        | $4.6(\pm2.3) \times 30$       | $5.8(\pm1.9) \times 30$                   | N/A                                |
| Synthetic-unequal | $3.7(\pm0.9) \times 30$        | $4.0(\pm1.6) \times 30$       | $2.5(\pm1.4) \times 30$                   | N/A                                |
| Adult             | $44.6(\pm23.3) \times 30$      | $56.6(\pm39.6) \times 30$     | $3785.0(\pm752.7) \times 10$              | $52.8(\pm61.4) \times 30$           |
| Bank              | $43.1(\pm27.1) \times 30$      | $70.1(\pm49.0) \times 30$     | $2835.4(\pm532.1) \times 5$               | $109.2(\pm65.0) \times 30$          |
| Census II         | $6423.5(\pm4072.2) \times 5$   | $7838.7(\pm6202.4) \times 5$  | $261.6(\pm1.3) \times 5$ (invalid results) | $127174.1(\pm13039.6) \times 3$     |

Table 11: Runtime in seconds for different algorithm-dataset combinations. The number after $\times$ indicates how many runs were executed. "Ncut Lipschitz" indicates the Fair Ncut algorithm with tuned (lower) Lipschitz values.

## F Kernel

### F.1 Computational complexity

In each iteration of the kernel-based clustering algorithm, a kernel matrix $\mathbf{K}$ is calculated, such that $[\mathbf{K}]_{ij} = \kappa(\mathbf{x}_i, \mathbf{x}_j)$. This operation is followed by calculating the distances from the points to each centre which is an operation of complexity $\mathcal{O}(N^2 K)$ when using the kernel matrix. Hence, the computational complexity of the implementation of the kernel-based clustering algorithm is $\mathcal{O}(N^2 KM)$, where $M$ is the maximum number of iterations. Note that, without the usage of the kernel matrix, the computational complexity of finding the distance between any point and a cluster centre is $\mathcal{O}(N^2)$ rather than $\mathcal{O}(N)$, implying that the total complexity of the algorithm would be $\mathcal{O}(N^3 KM)$. The lower computational complexity comes at the cost of a higher memory complexity of storing the kernel matrix.

### F.2 Kernel choice

Not any function $\kappa : X \times X \to \mathbb{R}$ describes a useful kernel. To allow the underlying function that is described by the kernel to map to an infinite dimensional space, the kernel $\kappa$ needs to satisfy Mercer's condition (Cortes and Vapnik, 1995), i.e. it must hold that

$$\iint \kappa(\mathbf{x}, \mathbf{y})g(\mathbf{x})g(\mathbf{y})d\mathbf{x}d\mathbf{y} > 0,$$

for all $g$ such that

$$\int g^2(\mathbf{x})d\mathbf{x} < \infty.$$

Three well-known kernels to satisfy Mercer's condition are the polynomial kernel, the Gaussian kernel, and the hyperbolic tangent kernel, which is why these kernels are the ones analysed in this work.

| Kernel type | Definition | Parameters |
|---|---|---|
| Polynomial | $\kappa(\mathbf{x}_i, \mathbf{x}_j) = (\mathbf{x}_i \cdot \mathbf{x}_j + b)^d.$ | $b, d$ |
| Gaussian | $\kappa(\mathbf{x}_i, \mathbf{x}_j) = \exp\left(-||\mathbf{x}_i - \mathbf{x}_j||^2/2\sigma^2\right)$ | $\sigma$ |
| Hyperbolic Tangent | $\kappa(\mathbf{x}_i, \mathbf{x}_j) = \tanh(a(\mathbf{x}_i \cdot \mathbf{x}_j) + b)$ | $a, b$ |

Table 12: Definition of three kernel types that satisfy Mercer's condition.

### F.3 Kernel results

| Dataset | SC | | Fairness error / Balance | |
|---|---|---|---|---|
| | VFC | VFC (kernel) | VFC | VFC (kernel) |
| **Polynomial Kernel ($b = d = 2$)** | | | | |
| Synthetic ($\lambda = 100$) | 0.649 | 0.649 | 0.00/1.00 | 0.00/1.00 |
| Synthetic-unequal ($\lambda = 100$) | 0.739 | 0.739 | 0.00/0.33 | 0.00/0.33 |
| Bank 2.5k ($\lambda = 100$) | 0.598 | 0.598 | 0.02/0.18 | 0.02/0.18 |
| **Radial Kernel ($\sigma = 2$)** | | | | |
| Synthetic ($\lambda = 100$) | 0.649 | 0.649 | 0.00/1.00 | 0.00/1.00 |
| Synthetic-unequal ($\lambda = 100$) | 0.739 | 0.739 | 0.00/0.33 | 0.00/0.33 |
| Bank 2.5k ($\lambda = 100$) | 0.598 | 0.598 | 0.02/0.18 | 0.02/0.18 |
| **Hyperbolic Tangent ($a = b = 2$)** | | | | |
| Synthetic ($\lambda = 100$) | 0.649 | 0.649 | 0.00/1.00 | 0.00/1.00 |
| Synthetic-unequal ($\lambda = 100$) | 0.739 | 0.739 | 0.00/0.33 | 0.00/0.33 |
| Bank 2.5k ($\lambda = 100$) | 0.598 | 0.598 | 0.02/0.18 | 0.02/0.18 |

Table 13: Comparison of the vanilla $K$-means, and kernel-based VFC algorithms using SC, fairness error and balance.

