# OpenReview forum: "Reproduction Study of Variational Fair Clustering"
_ML_Reproducibility_Challenge/2021/Fall — RC2021_

### Official Review · Reviewer_Xv3C · 2022-02-23
**Review for Reproduction Study of Variational Fair Clustering**

**Rating:** 7
**Confidence:** 3

**Review:**

Scope of reproducibility

The report presents clearly the scope of reproducibility and adheres to it.

Code

The code of the original author is re-used (with removing bugs). A hyperparameter search is
performed for the additional experiments. The code is submitted with a clear guidance.

Communication with original authors

A fair communication is established with the original authors to clarify the implementation
details.

Hyperparameter Search

The authors perform a turning the Lipschitz constant parameter

Discussion on results

The authors report clearly and discuss the results of both reproduction and replication parts.
The easy and difficult parts in the reproducing are well presented.

Recommendations for reproducibility

The authors give some recommendations to the original authors w.r.t correctness of the
parameter, Lipschitz value and errors in the original code.

Results beyond the paper

The author report the new results on new dataset, and a runtime of experiments is presented as well.

Overall organization and clarity

The report is well-written and has a good structure.

---

### Official Review · Reviewer_kcAp · 2022-02-26
**Comprehensive analysis on the reproducibility of a fair clustering algorithm**

**Rating:** 8
**Confidence:** 4

**Review:**

This work investigates the reproducibility of a subset of validation benchmarks for a recently proposed variational clustering approach. The authors include also additional analyses to evaluate the robustness and scalability of the method. The authors confirm overall promising performance of the suggested method, and find support for most of the key claims in the original work but also report notable limitations in reproducibility and reservations with respect to the overall support towards one of the major findings in the original paper. With help from the original authors and a detailed exploration of technical details, the authors could largely replicate the analyses presented in the original paper.

Quality & clarity: The scope is well described and followed throughout the report, although the additional experiments beyond the original paper could be more clearly mentioned in the scope. On the other hand, the reporting is comprehensive and provides detailed information on the analysis. Subsequently, the analysis includes a number of different data sets, experiments, parameters, and acronyms, making the reporting somewhat heavy to follow. The overall quality of the work is high, with relevant details are investigated and reported carefully.

Originality & significance: this is a detailed reproducibility report, although its scope covers also a subset of the analyses in the original work. The original source code could be used to a large extent but additional support from the original authors was essential for achieving full reproducibility. Some discrepancies between the code and original report were identified during this process. The work provides recommendations for improving reproducibility and suggests that a more versatile set of large data sets, kernels, and parameter exploration would be useful to assess the robustness of the new method.

Pros:
- The analysis is detailed and comprehensive, with clear conclusions and recommendations
- Some errors in the reported parameters are identified but in most cases resolving these do not impact the main results
- The report goes beyond the original work by including experiments with a new data set
- New opportunities to speed up calculation are identified

Cons:
- I did not find the source code for reproducibility analyses
- The level and versatility of details makes the main report a bit hard to follow
- I spotted occasional typos while reading; proofreading/spellcheck is recommended

---

### Meta-Review · Program_Chairs · 2022-04-09

**Recommendation:** Accept
**Confidence:** 5

**Metareview:**

The paper is a solid submission by the authors.  It is recommended for acceptance.

---

### Decision · Program_Chairs · 2022-04-09

**Decision:**

Accept

**Comment:**

Following the recommendation of reviewers and meta-reviewer, the paper is accepted for ML Reproducibility Challenge 2021, and will be published in the upcoming special edition of ReScience Journal.